# Hedgehog Pathway Inhibitors against Tumor Microenvironment

**DOI:** 10.3390/cells10113135

**Published:** 2021-11-12

**Authors:** Silpa Gampala, Jer-Yen Yang

**Affiliations:** 1Department of Pediatrics, Indiana University School of Medicine, Indianapolis, IN 46202, USA; gsiitm011@gmail.com; 2Research Center for Cancer Biology, Graduate Institute of Biomedical Sciences, College of Medicine, China Medical University, Taichung City 406040, Taiwan

**Keywords:** hedgehog pathway, cancer, HH pathway inhibitors, drug resistance, immunotherapy, tumor microenvironment

## Abstract

Targeting the hedgehog (HH) pathway to treat aggressive cancers of the brain, breast, pancreas, and prostate has been ongoing for decades. *Gli* gene amplifications have been long discovered within malignant glioma patients, and since then, inhibitors against HH pathway-associated molecules have successfully reached the clinical stage where several of them have been approved by the FDA. Albeit this success rate implies suitable progress, clinically used HH pathway inhibitors fail to treat patients with metastatic or recurrent disease. This is mainly due to heterogeneous tumor cells that have acquired resistance to the inhibitors along with the obstacle of effectively targeting the tumor microenvironment (TME). Severe side effects such as hyponatremia, diarrhea, fatigue, amenorrhea, nausea, hair loss, abnormal taste, and weight loss have also been reported. Furthermore, HH signaling is known to be involved in the regulation of immune cell maturation, angiogenesis, inflammation, and polarization of macrophages and myeloid-derived suppressor cells. It is critical to determine key mechanisms that can be targeted at different levels of tumor development and progression to address various clinical issues. Hence current research focus encompasses understanding how HH controls TME to develop TME altering and combinatorial targeting strategies. In this review, we aim to discuss the pros and cons of targeting HH signaling molecules, understand the mechanism involved in treatment resistance, reveal the role of the HH pathway in anti-tumor immune response, and explore the development of potential combination treatment of immune checkpoint inhibitors with HH pathway inhibitors to target HH-driven cancers.

## 1. Introduction

Challenges of targeting brain tumors include intrinsic immunosuppressive environment, lack of antigen targets, antigenic variability, and immune-restrictive site of the central nervous system [1]. Medulloblastoma (MB) is the most common malignant pediatric brain tumor with a 5-year survival rate of 40–80%, depending on the subtypes. WNT, SHH, group 3, and group 4 are the four major subtypes of MB [2], with their survival rates of 90%, 70%, 50%, and 50–90%, respectively [3,4,5]. Research on 100 most influential MB studies indicates that sonic hedgehog (SHH) pathway aberration is the main culprit for SHH-MB initiation and progression [6]. Apart from MB, basal cell carcinoma (BCC) is another cancer with various mutations of SHH pathway components [7,8,9,10,11,12]. SHH pathway is also extensively implicated in other malignancies, including estrogen receptor (ER+)-positive and triple-negative breast cancer (TNBC), for which overall survival and disease-free survival are 62% and 57%, respectively [13,14,15,16]. In addition to canonical SHH signaling, the non-canonical pathway is known to activate GLI1 such as the PIK3/AKT (phosphatidylinositol 3-kinase/protein kinase B), EGFR (epidermal growth factor receptor), TGF-β (transforming growth factor-beta), and NF-κB (nuclear factor kappa light chain enhancer of activated B cells) and those pathways are the main cause of treatment failure [17]. SHH signaling is also activated in the stem cells that enable adaptation to hypoxic conditions, thus promoting breast cancer metastasis [17,18,19,20,21,22,23].

## 2. Role of HH/SMO/GLI1 Signaling in Tumor Development

Primary cilia are present in almost every type of cell, including blood cells in eukaryotes [24]. Primary cilia participate in various cellular functions such as chemo-sensation, signal transduction, cell division, and differentiation [25]. They function through G-protein-coupled receptor (GPCR) signaling, calcium or other ion channels, and several signaling pathways, including the SHH pathway. Emerging literature indicates that primary cilia could promote or inhibit cancer progression depending on the cancer type. They are implicated in mediating paracellular signals between tumor cells and their microenvironment that control cancer growth, metastasis, and therapeutic responses. Cilia in cancer cells harbor cell growth and drug resistance phenotypes through governing a myriad of signaling systems such as SHH, WNT, NOTCH, PDGF, and other receptor tyrosine kinases (such as IGF1R, FGFR) [26,27,28,29].

In the absence of SHH ligand, the 12-transmembrane protein, PTCH1 inhibits the 7-transmembrane GPCR, namely smoothened (SMO), by preventing SMO from translocating into the primary cilia, and thereby blocks GLI transcription factors from moving to the nucleus, where GLI transactivates SHH target genes to promote cancer cell growth and metastasis [30,31]. On the contrary, binding of SHH to PTCH1 can promote internalization of PTCH1 and, in turn, SMO gets to undergo cilia localization to sequester SUFU (suppressor of fused) away from GLI-mediated inhibition, leading to activated gene transcription contributing to cell proliferation and metastasis [30,32] (Figure 1). Dysregulated activation of the HH pathway has been associated with a variety of cancers such as BCC, MB, breast, prostate, and lung cancers. This includes overexpression of HH ligand in tumor and tumor microenvironment (TME), loss of function mutation in PTCH1 and *SUFU*, *GLI2* amplifications, and gain of function mutation in SMO [10,31,33,34,35,36,37,38,39].

It has been shown that secretion of SHH ligand, cytokines, and chemokines and their dynamic crosstalk in an autocrine or paracrine manner in the TME can amplify SHH signaling [40]. Aberrant expression of SHH as well as non-canonical [24,41,42] SHH signaling results in TME remodeling, which in turn induces GLI1 upregulation in breast, pancreatic, and brain cancers (Figure 1) [41,42,43,44]. Additional to SHH signaling activity, multiple tumors interacting and influencing cells such as vascular cells, immune cells, astrocytes, microglia, stem cells, and even extracellular matrix can contribute to the advanced development of MB [45]. Moreover, the SHH enriched MB significantly increases gene expression of tumor-associated macrophages, thus resulting in an abundance of M2 subtype-tumor-associated macrophages, and patients with increased macrophage M2 show significantly worse prognosis [46,47,48].

In addition to this canonical function, the SHH pathway cross talks with other tumorigenic pathways such as MAPK, mTOR, AKT, and PI3K, thus driving cancer progression [46,47,49]. All these pathways enhance SHH target gene expression, including CCND1, FOXM1, VEGF, ABCG2, NRP2, among others [48]. GLI can also be regulated by TGFb1, BET proteins, PRMTs, and HDAC proteins [50,51,52,53,54]. For example, BET protein Brd4 binds to the promoter region of *GLI1* and *GLI2* to modulate their expression levels [18,55,56,57,58], and HDAC protein HDAC1 deacetylates Lys 518 of GLI1 protein to mediate transcriptional activation [59].

## 3. Hedgehog/GLI Pathway Inhibitors

Drug development faces many challenges even at a preclinical stage [60]. Most of the small molecules previously failed for clinical application mainly due to a lack of specificity and high toxicity that leads to severe side effects [61]. Further challenges arise at the clinical level where the developed molecule loses importance due to heterogeneity of the cancer cell with respect to expression of the small molecule target [62,63]. Adverse events such as hair loss, fatigue, taste alterations appear in most HH inhibitor-treated patients that lead to discontinuation of treatment in at least 30% of patients [64,65]. All these need to be taken into consideration while developing small molecule inhibitors. Several inhibitors have been successfully developed against HH pathway proteins that are currently in the clinic or clinical trials. However, all those may face complications due to continued acquired resistance in patients over time. Newer methods of targeting such as immunotherapy and efforts to circumvent adverse effects of chemotherapy using stem cell transplantation (NCT00002594) [66,67] are emerging but are quite expensive to afford [68].

Current inhibitors against HH signaling include 5E1 (against SHH); Cyclopamine, vismodegib, sonidegib (against SMO); ATO, Gant-58, Gant-61 (against GLI1/GLI2 transcription factor), among others. These molecules have been extensively reviewed, and hence only prominent among them are discussed below. A list of inhibitors and their targets are provided in the table (Table 1).


**SHH inhibitors**


Three inhibitors of SHH are currently under preclinical study, but none have reached the clinical stage. RU-SKI 43 is a dihydrothienopyridine derivative that inhibits HH acyltransferase responsible for N-terminal palmitoylation of SHH for its efficient signaling function [71]. The monoclonal antibody, 5E1, is another that prevents the binding of SHH from interacting with PTCH1 protein, thus blocking HH signaling [69]. Recently discovered small molecule (7_3d3 with IC50 of 0.4 ± 0.1 μM in cellular assays) against SHHN (sonic hedgehog N-terminal) warrants further investigation with high promise to target HH signaling [73].


**SMO Antagonists**


Small molecule SMO antagonists target HH signaling by binding to the pockets within the extracellular domain (ECD) and the transmembrane domain (TMD) of SMO [96]. Vismodegib was the first of such antagonists approved by the FDA in 2012 for the treatment of metastatic or locally advanced BCC [47]. Vismodegib binds to SMO to disrupt its structural conformation for sustained activation [75,77]. Since then, advances have been made for new SMO antagonists due to the development of resistance to vismodegib and also due to low responsiveness in some types of HH-driven cancers. The main cause was due to potential mutations within the drug-binding pockets of SMO [97,98]. Sonidegib (biphenyl carboxamide) that interacts with the residues from the extracellular tips of helices I, II, V, and VII of the SMO drug-binding pocket was then approved by the FDA in 2015 for BCC [81]. Among SMO antagonists, cyclopamine, which was the very first HH inhibitor, has been only used in preclinical studies and has failed clinically due to poor solubility, stability, and moderate activity [99], while TAK-441 and LEQ506 are in Phase 1 clinical trials, and taladegib (interacts with residues from extracellular loop 3 (ECL3) of SMO, including Q477, W480, E481 and F484 residues [79]), saridegib, XL139, glasdegib, as well as itraconazole, are in phase II clinical stages [100,101,102].


**GLI inhibitors**


Downstream inhibition of HH signaling makes therapy possible in the event of failure of upstream inhibition. In addition, vismodegib-resistant cancers exhibit hyperactivation of GLI protein and activity [103,104]. GLI proteins are transcription factors functioning as downstream effectors of both canonical and non-canonical HH signaling pathways. GLI1 and GLI2 are activated when dissociated from their negative regulator SUFU [105]. Thus, the non-canonical activation of these GLI proteins contributes to resistance to SMO inhibitors. Some of those pathways include RAS-RAF-MEK-ERK (rat sarcoma Raf proto-oncogene, serine/threonine kinase-mitogen-activated protein kinase kinase-MAPK extracellular signal-regulated kinase), AMPK-mTOR-S6K [78,106], TGFβ, and others [107]. Gant-58 inhibits *GLI1* transcription, and Gant-61 inhibits both GLI1 and GLI2 activity. HPI-1, HPI-2, HPI-3, and HPI-4 are HH pathway inhibitors that work through an unknown mechanism to block GLI protein activity. However, none of these inhibitors have been employed in any clinical trials to date [91].

## 4. Resistance Mechanisms to HH Inhibitors

The first report of resistance to vismodegib was published in 2009 when an MB patient treated with vismodegib relapsed and died [108]. This case revealed mutations in the *SMO* gene, and since then, several resistance contributing mutations in critical HH pathway genes have been reported. Following, we summarize and discuss mutations in major HH components that contribute to resistance.


**SMO mutations**


*SMO* is the most mutated gene of the HH pathway in BCC and MB patients. These mutations result in de novo resistance or acquired resistance). G497W, D473Y, D473H mutations in SMO have been analyzed and described by Yauch RL et al., Pricl S et al., demonstrated that G497W resulted in partial obstruction of drug entry site, while D473Y affected the drug binding, thus conferring primary and secondary vismodegib resistance, respectively [98,109]. Neither vismodegib nor sonidegib is effective against D473H SMO mutants [110]. In BCCs, several mutations within the transmembrane (TM-1 or 2) domain of SMO were identified, but H231R, W281C, V321, I408V, D473G, C469Y, and Q477E displayed impaired vismodegib binding [96], whereas SMO missense mutations L225R, N223D, S391N, D388N, and G457S severely decreased sonidegib potency [111]. Sharpe HJ et al. also reported SMO mutations within (W281C, V321M, I408V, C469Y) and outside (T241M, A459V, L412F, S533N, and W535L are outside the drug-binding pocket, DBP) the DBP to be responsible for resistance in BCC patients [112]. Jain S et al. found that Q476 and D473 mutations within the DBP prevent sonidegib binding, whereas those at S533 and W535 block sonidegib’s access to the DBP [81].


**HH and GLI amplifications**


Hyperactivation of GLI factors is the main culprit of chemoresistance or radiation resistance in glioma, pancreatic, prostate, and breast cancer patients [50]. *GLI2* amplifications were previously reported to be associated with HH inhibitor resistance in MB and BCCs [99,113,114]. Overproduction of the HH ligands constitutively activates the pathway in an autocrine manner, thus leading to a decrease in the efficiency of HH inhibitors over time [115].

Accumulated reports indicate the only possible way of combating drug resistance is using combination therapy [110]. A drug holiday is one way to combat adverse side effects of drugs where patients are intermittently treated with inhibitors and then left without treatment to recover from the side effects [113]. Repurposing clinically investigated drugs was shown to be able to overcome classical HH inhibitor resistance. For example, a series of BRD4 inhibitors based on AbbVie’s phase I clinical pan-BET inhibitor 2 (ABBV-075) yielded Compound 25 to be a safe, tolerant, and high potent GLI inhibitor both in vivo and in vitro [55]. Interestingly, recent studies showed that HH signaling was also associated with tumor immunosuppression [116]. Patients could have elevated responses to HH inhibitors if HH-mediated tumor immunosuppression is better exploited and targeted.

## 5. Hedgehog Signaling Suppresses Anti-Tumor Immune Response

HH/GLI1 signaling regulates immune checkpoint modulators such as PD1, CTLA-4, TIM3, LAG3, TIGIT, and IL-10 in exhausted T cells as well as PD-L1/2, CD80/86, OX40L, CD137L, IDO, and CCL22 in cancer cells [95,114,117,118]. HH signaling induces PD-L1 expression in cancer cells mostly mediated by cytokines such as IFN-gamma that suppress the activation of cytotoxic T-lymphocytes [119,120]. Additional upregulation of PD-L1, PD-L2, TIGIT, TIM3, and CD226 was reported in BCC-like skin tumors where the TME is enriched in T-cell populations overexpressing PD-1 [121]. Regulatory T cells limit auto-immune response, inflammation response, as well as anti-tumor immunity.

Studies show that the HH pathway is involved in remodeling the tumor microenvironment (TME), thus regulating anti-tumor immunity [116]. TME consists of different non-neoplastic cells such as cancer-associated fibroblasts (CAF), immune cells, endothelial cells, and neurons that communicate with tumor cells [122]. This crosstalk promotes tumor progression via modulation of TME plasticity, immune suppression, metastasis, etc. TME immune-suppressive cells include M2 macrophages (TAMs), Treg cells, tumor-associated neutrophils (N2 or TANs), and myeloid-derived suppressor cells (MDSC) [80] (Figure 1). Tumor-associated macrophages (TAMs) expressing M2 (alternatively activated)-like phenotype imitate type II T-helper cells that express interleukin (IL)-4, IL-5, IL-6, IL-13, and IL-10 to suppress anti-tumor immune response [120,121]. CAF can produce tumor-promoting cytokines such as CXCL12, IL-6, HIF1a, and TGF-b2 [123], and monocytic MDSCs can promote tumor epithelial to mesenchymal transition (EMT) [80].

It was also shown that the inhibition of HH signaling reprogramed the dysfunctional immune microenvironment in breast cancer [114]. Petty et al. demonstrated that conditional knockout of SMO in myeloid cells such as macrophages, monocytes, and granulocytes using LysMcre+Smo^fl/fl^ mice interfered with tumor growth by disrupting the M2 TAM polarization [124]. Furthermore, inhibition of the HH pathway using vismodegib and sonidegib reduced the number of cilia in BCC as well upregulated expression of MHC class I, attracted MHC class II+ T cells, CD4+ T cells, and CD8+ T cells into the TME [125]. It was proposed that HH signaling plays a significant role in reducing the strength of T-cell receptors in mature peripheral T cells [126]. Immune-suppressive cytokines such as TSLP (thymic stromal lymphopoietin), TGFβ, IL-10, and INOS were often found to be elevated in HH-hyperactivated cancers [121].

## 6. Immune Checkpoint Blockade against Hedgehog Prominent Cancers

The anti-tumor immunity involves T-cell generation and activation, infiltration of T cells into TME, and successful T cells targeting tumor cells for destruction. Tumor cells express immune checkpoint proteins, thus adopting immune evasion recognition mechanisms [80]. Advances in immunotherapy brought breakthroughs for many aggressive and non-responsive cancers. Immunotherapy works to enhance the T-cell’s ability to recognize antigens presented by MHC-I proteins present on tumors [127]. Immunotherapeutic modalities employed against different cancers involve cancer vaccines, oncolytic viruses, checkpoint inhibitors, natural killer cells, and CAR-T cell therapy [1]. However, not every patient does not respond to immunotherapy, and the clinical results are often not significant. Further, responders develop primary and acquired resistance due to the involvement of metabolic, inflammatory, and vasculatory mechanisms within the tumor as well as in the TME [85].

Evaluation of two preclinical models (SHH-dependent and SHH-independent subtypes) of MB showed differential efficacies of anti-CTLA-4 and anti-PD-1 antibodies, indicating that immunologic differences within the TME may determine response to immune checkpoint inhibitors (ICIs) [128]. Importantly, PD-L1 expression was particularly high in the SHH subtype of MB cells, while differential expression of PD-L1 was attributed to differential tumor immune response [129]. It was shown that HH signaling induced PD-L1 expression and inactivated effector T cells in gastric cancer cells to facilitate their proliferation, while inhibition of HH signaling reversed GLI2-induced tolerance [120].

A recent preclinical report from D. Orlando et al. identified the expression of PRAME (an antigen preferentially expressed in melanoma, SLL, serine leucine leucine) in 82% of 60 MB patient biopsies and showed that this antigen can be targeted using genetically modified T cells (SLL TCR T cells with inducible caspase-9) [130]. Plasma and tumor tissue samples from immune checkpoint inhibitor (pembrolizumab and nivolumab)-treated NSCLC patients revealed elevated levels of Wnt and SHH in plasma as well as increased GLI2 levels, suggesting HH and Wnt activation was correlated with immune therapy resistance [80,131]. BCC patients exhibit greater mutational burden than any other cancer reported and thereby makes it a better target for immunotherapy [112,132,133] since high mutation burden (TMB-H, i.e., mutations/Mb) enables the expression of immunogenic neoantigens that can be recognized by T cells, thus increasing the efficacy of immunotherapy [134]. Vismodegib- and sonidegib-treated BCC patients exhibit increased TILs as well as higher MHC-I expression [125]. Another study reported increased tumor response to pembrolizumab (anti-PD-1) treatment in metastatic PDL1 (+) BCC patients who previously had been treated with HH targeting therapy [135]. Nivolumab (anti-PD-1) combined with ipilimumab treatment in locally advanced BCC (laBCC) and metastatic BCC (mBCC) patients is currently being investigated [64]. These studies highlight the importance of HH signaling for an immunosuppressive tumor environment and also implicate the potential of combining immune checkpoint inhibitors and HH inhibitors for more effective cancer therapy.

## 7. Combining HH Inhibitors and ICI Inhibitors

As described in the previous sections, both canonical and non-canonical activation of HH downstream effectors contributes to drug resistance and cancer relapse. Additionally, HH signaling in TME and its contribution to immune suppression are key factors for designing therapeutic regimens to combat drug resistance. Targeting the HH pathway does not seem to work in metastatic or recurrent cancers due to various factors, including HH target gene mutations, gene amplifications, immune-suppressive TME, and others [36]. Many HH inhibitors are still under clinical trial with adaptive resistance (e.g., SMO inhibitors). Hence, understanding drug resistance and combining HH inhibitors with immune therapy need to be explored pronto (Figure 2).

An exploratory study conducted by Leandro M. Colli et al. using cancer genomic data sets for somatic mutation profiles indicated that a significant proportion of SMO mutated patients could be benefited from a combination of immunotherapy with targeted therapy considering the mutational burden [136]. Petty A J et al. reported that vismodegib combined with anti-PD-1 antibody resulted in a synergistic reduction in liver tumors in mice (Hepa1-6 and LLC-1 tumors) by reversing M2 to M1 TAMs and increasing CD8+ T-cell trafficking into the TME [124]. A limited size study conducted by Dr. Anne Lynn S. Chang (NCT02690948) showed a 44% vs. 29% overall response rate compared between pembrolizumab-treated patients (*n* = 9) and pembrolizumab + vismodegib combination-treated patients (*n* = 7), suggesting that immunotherapy may work better than combination. However, one-year progression-free survival probability favored combination therapy (62% vs. 83%) [94,95].

## 8. Conclusions and Future Direction

HH signaling controls embryogenesis and organ development. HH signaling has been studied extensively in tumor cells, whereas the impact of HH signaling on the immune TME is a newly explored territory [114]. It is shown that SHH promotes macrophage M2 polarization and reduces effector CD8+ T-cell recruitment to the tumor [124,137]. A high percentage of M2 tumor-associated macrophage (TAM) is correlated with poor patient outcomes [138]. Previous efforts employed to eliminate the TAMs in breast and lung cancer have yielded successful clinical outcomes [139,140]. This is, in part, due to the fact that immunotherapeutic strategies are successful for highly heterogeneous tumor types with high T-cell infiltration while remaining less effective for tumor types that have limited T-cell infiltration. Hence, inhibition of HH signaling could present dual benefits for directly targeting tumor cells and re-configuring the tumor immune microenvironment to an immune active state.

Emerging evidence shows that immunotherapeutic strategies can contribute to better cancer patient survival [141]. Pediatric MB is the most frequently diagnosed brain tumor in children [142]. The prognosis is mainly dependent on the molecular subtype of the tumor, although therapeutic strategies are limited to conventional radiation therapy, chemotherapy, and surgery [3,143]. Due to severe neurological side effects [144], strategies such as immunotherapy are being actively investigated. Castriconi et al. showed for the first time that NK cells can kill MB cells in vitro, which opened up a new avenue to study the potential of NK cell-based immunotherapy in MB [145]. Tumor-infiltrating lymphocytes (TILs) were reported to be detected in pediatric MB lesions [146]. The main TIL subsets are CD3+, CD8+T cells, which had predominantly a perivascular and intratumoral infiltration pattern. The TILs were barely activated given the low percentage of granzyme B (GrB)- and PD1-positive cells. It has been hypothesized that pediatric and embryonic tumors are not immunogenic, and therefore immunotherapeutic interventions have limited success compared to non-small cell lung cancer or melanoma [147]. Clinical studies with GBM patients revealed that the majority of the patients had tumor cells expressing PD-L1, and activation of the PD1/PD-L1 axis is associated with poor prognosis [148]. CD8+ T cells are enriched in murine medulloblastoma, which is often PD1-positive; as a result, administration of PD1 blocking antibodies can have beneficial survival effects [128]. A study compared between several pediatric tumors revealed that GBM, neuroblastoma, as well as the embryonic atypical teratoid/rhabdoid tumor, had increased number of TILs along with increased expression of PD-L1 [149], explaining the influx of TILs failed to improve the overall survival of MB patients [150,151].

Another report indicated that SHH-MB tumors contained significantly increased infiltrating dendritic cells (DC), T cells, and myeloid cells in the TME comparison to group 3 MB tumors. High percentages of PD-1+ CD8 T cells were identified in group 3 MB tumors; therefore, in vivo blockade of PD-1 expressing lymphocyte population showed significant anti-tumor effects group 3 MB-bearing animals, which did not work for SHH-MB animals. This study suggests that different MB subgroups have distinct immune profiles that may require different immunotherapeutic targeting strategies [128].

Although immunotherapy has shown promising results for cancer treatment, several cancer types, including brain tumors, show limited response to these treatments. Future studies that investigate the key molecular mechanism involving low immune response in the cold tumor are needed. Research shall focus on cell-cell interaction in the TME, reveal how different cell populations, such as immune cells, fibroblast, etc., crosstalk and interact with the tumor cells will provide a more complete picture for the development of effective targeting strategies.

## Figures and Tables

**Figure 1 cells-10-03135-f001:**
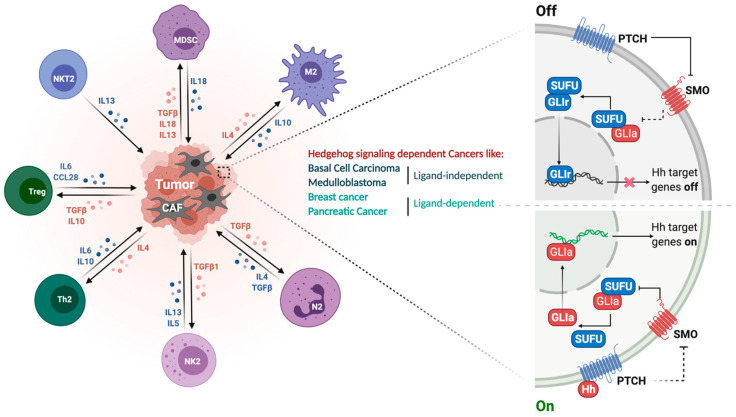
SHH signaling pathway and its immunosuppressive tumor microenvironment. Right panel represents the ON-OFF signaling of the HH pathway. In the absence of HH ligand, Ptch1 inhibits surface localization of Smo, leaving Sufu free to bind to Gli, thus repressing it and preventing HH target gene expression. In the presence of HH ligand, Gli is released from Sufu to translocate into the nucleus, thus activating HH target genes. Left panel elaborates different immunosuppressive cell types infiltrated into the tumor microenvironment and the associated cytokines and growth factors. Oncogenic HH signaling recruits immune cells such as tumor-associated macrophages (TAMs), immune-suppressive myeloid-derived suppressor cells (MDSCs) for immune modulation.

**Figure 2 cells-10-03135-f002:**
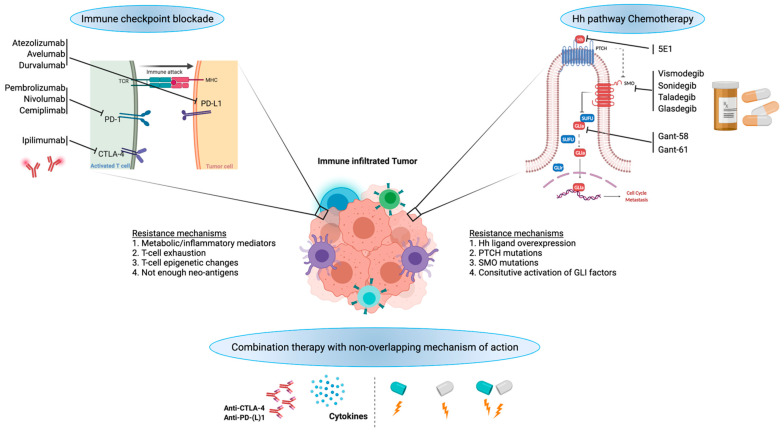
Potential for combination of HH signaling inhibition with immunotherapy.

**Table 1 cells-10-03135-t001:** HH pathway inhibitors in preclinical or clinical phases and the immune checkpoint inhibitors for potential combination studies.

HH Pathway Inhibitors	Target	Reference	Immune Checkpoint Inhibitors	Target	Reference
5E1 monoclonal antibody	SHH	[69]	Atezolizumab	PD-L1	[70]
RS-U 43	SHH	[71]	Avelumab	PD-L1	[72]
7_3d3	SHH	[73]	Durvalumab	PD-L1	[1]
Robotnikinin	SHH	[28]	Dostarlimab	PD-1	[74]
Vismodegib/GDC-0449	SMO	[75,76,77,78]	Cemiplimab	PD-1	[64]
Glasdegib (PF-04449913)	SMO	[79]	Nivolumab	PD-1	[80]
Erismodegib/LDE225/sonidegib	SMO	[76,81]	Pembrolizumab	PD-1	[80]
Taladegib (LY2940680)	SMO	[79]	Ipilimumab	CTLA-4	[80]
SANT-1	SMO	[69]	Aldesleukin	IL-1/IL-2R	[1]
LEQ506	SMO	[82]	Interferon alpha-2a	IFNAR1/2	[83]
BMS-833923 (XL-139)	SMO	[84]	Interferon alpha-2b	IFNAR1/2	[83]
Saridegib/patidegib/IPI-926	SMO	[84]	PegIFN alpha-2b	IFNAR1	[83]
Itraconazole	SMO	[18,68,85]	Imiquimod	TLR7	[86]
CUR61414	SMO	[87]	Poly ICLC	TLR3	
ALLO-1 and 2	SMO	[88]	Pexidartinib	KIT, CSF1R, and FLT3	[89]
TAK-441	SMO	[48]	Tremelimumab	CTLA-4	[80]
ATO (arsenic trioxide)	GLI	[90]	Dostarlimab	PD-1	[74]
GANT-61	GLI	[69]	Cemiplimab	PD-1	[64]
GANT-58	GLI	[69]	Nivolumab	PD-1	[80]
HPI-1 (HH pathway inhibitor)	GLI	[91]			
Sirolimus	mTOR	[92]			
PF-4708671	S6K1	[24]			
PSI (PKC pseudosubstrate inhibitor)	aPKC	[93]			
Combination					
**Inhibitor**		**Target**		**Reference**	
Vismodegib + pembrolizumab		SHH + PD-1		[94,95]	

## Data Availability

Not applicable.

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
