# Peer review of "Hedgehog Pathway Inhibitors against Tumor Microenvironment"

_cells, 2021, doi:10.3390/cells10113135_

Round 1
Reviewer 1 Report
The manuscript entitled “Revealing Hedgehog Pathway Inhibitors in Drug-resistance and Tumor Microenvironment” submitted by Silpa Gampala and Jer-Yen Yang is a review of the role of the Hh signaling in tumor and in the tumor microenvironment immune response, of the effect of Hh pathway inhibitors and of immune checkpoint inhibitors on cancer treatment. The authors conclude their review by discussing the potential interest in combining Hh pathway inhibitors and immune checkpoint inhibitors for Hh dependent cancers treatment, which is interesting.
Here are my remarks:
Line 27: Replace “TME” with « tumor microenvironment (TME)” the first citing time.
Line 32: As MB is not the only cancer and Smo inhibitors are not the only Hh pathway inhibitors presented in this review, I suggest replacing “Medulloblastoma” and “Smoothened inhibitor” by “cancer” and “Hh pathway inhibitors” respectively in the keywords.
Line 49 : Replace « They function through G- 48 protein-coupled receptor (GPCR) signaling, calcium or other ion channels, and Hh signaling etc.” by something like “They function through G-protein coupled receptor (GPCR) signaling, calcium or other ion channels, and several signaling pathway including the Hh pathway”.
From lines 55 to 59: the authors describe the Hh signaling in healthy cells. They should give a ref to a recent review on Hh pathway signaling here. A sentence from line 54 “Dysregulated activation of Hh pathway has been associated with a variety of cancers like BCC, MB, breast, prostate, and lung cancers [26].” Would be better in line 60 after the normal Hh signaling description.
Line 55: Authors cite ref 26: Higgins, M.; Obaidi, I.; McMorrow, T. Primary cilia and their role in cancer. Oncol Lett 2019, 17, 3041-3047, doi:10.3892/ol.2019.9942. which does not seem to correspond. Please check. Ref 27 and 28 are more appropriate.
Line 60: Authors wrote “Deregulation of Hh pathway in cancers includes overexpression of Hh ligand in tumor microenvironment (TME),” . Overexpression of Hh ligand can also happen in the tumors (autocrine Hh signaling to correspond to type II cancers link to Hh signaling.
Figure 1 is not very clear and correct:
- In MB and BCC: overactivation of Hh signaling is due to Ptch1 mutations that inactivate Ptch1 and make Smo constitutively active.
- In other cancers such as breast, lung, prostate, pancreatic … Hh pathway is overactivated by the overexpression of Shh notably.
Authors should present here the different ways of Hh signaling activation in the different cancers cited (MB, BCC, breast and pancreatic cancers).
Line 90: The inhibitors cited by the authors are not all “clinical inhibitors” like cyclopamine. Replace “clinical inhibitors” with “inhibitors”.
Line 103: Replace “confirmation” with “conformation”
Line 132: The sentence “G497W, D473Y, D473H mutations in SMO have been analyzed and described by Yauch RL et al. and Pricl S et al. demonstrated that G497W resulted in…” is not correct.
Line 137: The sentence “Sharpe HJ et al. also reported SMO mutations within (W281C, V321M, I408V, C469Y) and outside (T241M, A459V, L412F, S533N, and W535L are outside the drug binding pocket, DBP) the DBP that can be responsible for resistance mechanisms in BCC patients” is not correct.
Line 142: Do the Ptch1 mutations reported here link to the resistance to Hh pathway inhibitors? This is not obvious to me. If this is not the case, this part should not be in the section “Resistance mechanisms in Hh inhibitors “.
Same remark for SUFU mutations.
Line 276: Table 1: Did the immune checkpoint inhibitors In the same line as Hh pathway inhibitors have been tested in combination? If this is the case, it would be good to indicate it in the Table legend. If this is not the case, it would be better to present immune checkpoint inhibitors in a Table 2.
Author Response
The manuscript entitled “Revealing Hedgehog Pathway Inhibitors in Drug-resistance and Tumor Microenvironment” submitted by Silpa Gampala and Jer-Yen Yang is a review of the role of the Hh signaling in tumor and in the tumor microenvironment immune response, of the effect of Hh pathway inhibitors and of immune checkpoint inhibitors on cancer treatment. The authors conclude their review by discussing the potential interest in combining Hh pathway inhibitors and immune checkpoint inhibitors for Hh dependent cancers treatment, which is interesting.
Line 27: Replace “TME” with « tumor microenvironment (TME)” the first citing time.
Response: We thank the reviewer for pointing this. The correction has been made.
Line 32: As MB is not the only cancer and Smo inhibitors are not the only Hh pathway inhibitors presented in this review, I suggest replacing “Medulloblastoma” and “Smoothened inhibitor” by “cancer” and “Hh pathway inhibitors” respectively in the keywords.
Response: We have changed the Keywords as suggested.
Line 49 : Replace « They function through G- 48 protein-coupled receptor (GPCR) signaling, calcium or other ion channels, and Hh signaling etc.” by something like “They function through G-protein coupled receptor (GPCR) signaling, calcium or other ion channels, and several signaling pathway including the Hh pathway”.
Response: The sentence has been replaced.
From lines 55 to 59: the authors describe the Hh signaling in healthy cells. They should give a ref to a recent review on Hh pathway signaling here. A sentence from line 54 “Dysregulated activation of Hh pathway has been associated with a variety of cancers like BCC, MB, breast, prostate, and lung cancers [26].” Would be better in line 60 after the normal Hh signaling description.
Response: This paragraph has been rearranged and new references were added.
Line 55: Authors cite ref 26: Higgins, M.; Obaidi, I.; McMorrow, T. Primary cilia and their role in cancer. Oncol Lett 2019, 17, 3041-3047, doi:10.3892/ol.2019.9942. which does not seem to correspond. Please check. Ref 27 and 28 are more appropriate.
Response: Corrections were made and highlighted as track changes.
Line 60: Authors wrote “Deregulation of Hh pathway in cancers includes overexpression of Hh ligand in tumor microenvironment (TME),” . Overexpression of Hh ligand can also happen in the tumors (autocrine Hh signaling to correspond to type II cancers link to Hh signaling.
Response: We thank the reviewer for pointing out this. The sentence has been corrected.
Figure 1 is not very clear and correct:
- In MB and BCC: overactivation of Hh signaling is due to Ptch1 mutations that inactivate Ptch1 and make Smo constitutively active.
- In other cancers such as breast, lung, prostate, pancreatic … Hh pathway is overactivated by the overexpression of Shh notably.
Authors should present here the different ways of Hh signaling activation in the different cancers cited (MB, BCC, breast and pancreatic cancers).
Response: Figure 1 is now modified to reflect the corrections.
Line 90: The inhibitors cited by the authors are not all “clinical inhibitors” like cyclopamine. Replace “clinical inhibitors” with “inhibitors”.
Response: The authors thank the reviewer’s careful examination. We apologize for overlooking this and we have corrected the error.
Line 103: Replace “confirmation” with “conformation”
Response: “confirmation” has been replaced with “conformation”.
Line 132: The sentence “G497W, D473Y, D473H mutations in SMO have been analyzed and described by Yauch RL et al. and Pricl S et al. demonstrated that G497W resulted in…” is not correct.
Response: We thank the reviewer for pointing this. The sentence has been corrected.
Line 137: The sentence “Sharpe HJ et al. also reported SMO mutations within (W281C, V321M, I408V, C469Y) and outside (T241M, A459V, L412F, S533N, and W535L are outside the drug binding pocket, DBP) the DBP that can be responsible for resistance mechanisms in BCC patients” is not correct.
Response: The sentence has been corrected as recommended.
Line 142: Do the Ptch1 mutations reported here link to the resistance to Hh pathway inhibitors? This is not obvious to me. If this is not the case, this part should not be in the section “Resistance mechanisms in Hh inhibitors “.
Response: Thanks for pointing out this question. This is true that PTCH1 mutations have not been reported to play a role in resistance mechanisms. This section has been removed.
Same remark for SUFU mutations.
Response: We thank the reviewer for raising this question. SUFU mutations have been implicated in resistance even though a thorough research has not been conducted. Reference for this has been added in this section.
Line 276: Table 1: Did the immune checkpoint inhibitors In the same line as Hh pathway inhibitors have been tested in combination? If this is the case, it would be good to indicate it in the Table legend. If this is not the case, it would be better to present immune checkpoint inhibitors in a Table 2.
Response: Not all checkpoint inhibitors have been tested in combination with HH inhibitors. Only PD-1 Antibody (Pembrolizumab) with Vismodegib has been tested as reported in the manuscript. Follow by reviewer’s suggestion, an additional section has been added to the table showing existing combination regimen.
Reviewer 2 Report
This review described the current understanding of the mechanism involved in treatment resistance of Hedgehog (Hh) pathway inhibitors and discussed development of potential combination treatment of immune checkpoint inhibitors with Hh inhibitors to target Hh-driven cancers. I think thet this review is well written and easy to understand. We believe this review will be of interest to many researchers involved in cancer research.
My specific comments are as follows.
1) In line 142-174, I understood that the PTCH1 mutation was not related to the effect of SMO inhibitors, but if so, the reference paper should be cited and described.
2) In line 172-173, the authors should describe exactly what "control" means. Moreover, please insert appropriate reference(s) in this sentence.
3) In line 182-184, the authors should insert appropriate reference(s) in this sentence.
4) The authors should write the explanation in Figure 1 in detail.
Author Response
This review described the current understanding of the mechanism involved in treatment resistance of Hedgehog (Hh) pathway inhibitors and discussed development of potential combination treatment of immune checkpoint inhibitors with Hh inhibitors to target Hh-driven cancers. I think thet this review is well written and easy to understand. We believe this review will be of interest to many researchers involved in cancer research.
1) In line 142-174, I understood that the PTCH1 mutation was not related to the effect of SMO inhibitors, but if so, the reference paper should be cited and described.
Response: We thank the reviewer’s suggestion. This section has been revised to remove PTCH1 mutations.
2) In line 172-173, the authors should describe exactly what "control" means. Moreover, please insert appropriate reference(s) in this sentence.
Response: We thank the reviewer’s careful examination. The sentence has now been modified and appropriate references were added.
3) In line 182-184, the authors should insert appropriate reference(s) in this sentence.
Response: The authors thank for the reviewer’s suggestion. Appropriate references have been added.
4) The authors should write the explanation in Figure 1 in detail.
Response: Figure 1 has been modified and the description reflects the figure details.
Reviewer 3 Report
the manuscript by Gampala and Yang reviews the status of Hedgehog pathway inhibitors in drug resistance and tumor microenvironment
The topic is of great interest, however, it is not well addressed.
A clear focus is missing, the reading results confused, too many aspects of the Hedgehog pathway are not explained well (i.e. line 63: “type II non-canonical Shh signaling 63 result in TME remodeling” without any mention of canonical nor non-canonical pathways)
The introduction is misleading, the role of HH in Medulloblastoma is far more complicated than what is reported (subgroups are just mentioned at the end of the manuscript). The role of HH in other solid tumors is confused and confusing.
Beyond SMO mutations, it is not clear the role of mutations in other molecules in therapy or resistance or tumor maintenance or tumor development?
The section on HH and tumor immunity seems slightly more clear, so my suggestion is that the authors rethink the review with that focus, perform a systematic review, such as using PRISMA method, that can really help the reader improving the knowledge.
For these reasons, I think the manuscript is not suitable for being considered in this form.
Author Response
the manuscript by Gampala and Yang reviews the status of Hedgehog pathway inhibitors in drug resistance and tumor microenvironment
The topic is of great interest, however, it is not well addressed.
A clear focus is missing, the reading results confused, too many aspects of the Hedgehog pathway are not explained well (i.e. line 63: “type II non-canonical Shh signaling 63 result in TME remodeling” without any mention of canonical nor non-canonical pathways)
Response: We appreciate the reviewer’s concern. This section has been modified including Figure 1. Non-canonical Hedgehog signals are classified as Type 1 which stem from PTCH and Type II that stem from SMO as referred by Raffaele Teperino et al (PMID: 24862854). We did not go into details as this was reviewed previously. For this reason, we included appropriate references now.
The introduction is misleading, the role of HH in Medulloblastoma is far more complicated than what is reported (subgroups are just mentioned at the end of the manuscript). The role of HH in other solid tumors is confused and confusing.
Response: We the reviewer’s comment. We now included in the introduction, MB subtypes and their survival rates as well as further description about breast cancer that is crucial to this review.
Beyond SMO mutations, it is not clear the role of mutations in other molecules in therapy or resistance or tumor maintenance or tumor development?
Response: We thank the reviewer for raising this concern. So far only SMO inhibitors are used in the treatment of cancers and other molecular mutations are not yet clear.
The section on HH and tumor immunity seems slightly more clear, so my suggestion is that the authors rethink the review with that focus, perform a systematic review, such as using PRISMA method, that can really help the reader improving the knowledge.
Response: We thank the reviewer’s suggestion. Since this is the newly developing field, we like to give a brief overview of HH inhibitor and TME and immune therapy in Hh-driven cancers. We believe after this more studies in HH and immune therapy will be reported. By the time, it will be good to have PRISMA to summarize the new finding.
For these reasons, I think the manuscript is not suitable for being considered in this form.
Reviewer 4 Report
This review manuscript by Gampala and Yang try to summarize the recent findings in the role that SHH signaling plays in controlling tumor microenvironment. Unfortunately, I feel that the present topic is poorly reviewed and I don’t feel the manuscript should be accepted in the present form. Here I list a few of issues I found.
Line 37- Define CNS
Line 38- “Despite a decent survival rate, the disease remission rate is very high”. This sentence makes no sense.
Line 39 – “Research on 100 most influential MB studies indicates that hedgehog (Hh) pathway aberration is the main culprit for MB initiation and progression” This is only true for SHH MB subtype (30% of the cases). The cell of origin of the rest of subtypes is not SHH driven.
Line 46- In general, Hh should be named SHH or HH when referring a protein/pathway. People in medulloblastoma field like to call it SHH and not HH. Sometimes you do Shh, other SHH, other Hh… please use one. Shh/Hh is a mouse gene.
Line 60 – “Deregulation of Hh pathway in cancers includes overexpression of Hh ligand in tumor microenvironment (TME), loss of function mutation in PTCH1 and SUFU, and gain of function mutation in SMO” Should include GLI2 amplifications.
Line 63 – Can authors explain what is type II here: “type II non-canonical Shh signaling…”? I never heard of type II SHH signaling.
Line 65 – “Brain tumors consist of tumors,” I guess you are missing the word cells after tumors, and should be singular in this case.
Line 67- ‘The SHH enriched MB significantly increases gene expression of tumor-associated macrophages. Tumor- associated macrophages (M2 subtype) are abundantly present in SHH MBs.” You are saying two times the same. Just one sentence would be enough.
Line 72 – “GLI can also be activated by upstream regulators, such as TGFb1, BET proteins, PRMTs, and HDAC proteins” BET and HDAC proteins are required for the transcription of Gli1. They are not activators of the pathway, but downstream regulators.
Line 73 – [53,54]: These manuscripts are not the original ones describing the regulation of the gli1 promoter by Brd4. The original ones were published years before. PMID: 25355313, PMID: 24973920
Fig 1 legend: “ON-OFF signaling of the Hh pathway” I don’t see any “on” “off”. Please do dual panels showing the on and the off state. You are only showing the on.
Line 88 – “Newer methods of targeting such as 88 immunotherapy and stem cell transplantation are emerging but are quite expensive to afford [56]”. The reference is not the right one. Also, can authors explain how a SHH tumor would be cured by using stem cel transplantation…. ?
Line 90 – “Current clinical inhibitors against Hh signaling include 5E1 (against Shh); Cyclopamine, Vismodegib, Sonidegib (against 90 SMO); ATO, Gant-58, Gant-61 (against GLI1 transcription factor) among others.” The only clinical inhibitors from this list are vismodegib, sonidegib and ATO. The rest are pre-clinical. Authors seem not to know there is a big difference.
Line 112 – “Adverse events like hair loss, fatigue, taste alterations appear in most Hh inhibitor treated patients that leads to discontinuation of treatment in at least 30% of patients, but so far, no treatment related deaths have been reported in clinical trial”. You start the GLI inhibitor section by talking about adverse effects in the clinic to GLI inhibitors??? I think you mean SMO inhibitors. If so this text doesn’t belong here. Remember besides ATO, no GLI inhibitor was tested in clinical trials.
Line 114- “Downstream inhibition of Hh signaling makes it possible to in the event of failure of upstream inhibition.” I cannot understand this sentence. Please fix.
Line 122- “A direct GLI inhibitor, Arsenic trioxide (ATO), is the only one that has been approved by the FDA for treating acute promyelocytic leukemia (77)” In the referenced paper GLI in never mentioned. ATO is FDA approved for this type of leukemia, but not as a GLI inhibitor. They even have a table in the manuscript with all tentative ATO targets, and GLI or SHH are not included.
Line 127/8- “This case revealed various SMO gene mutations in Vismodegib-resistant MB. PTCH1 has also been reported to be mutated thus resulting in Hh activation in majority of BCC and MB.” These 2 sentences make no sense. If you are talking about downstream of SMO activation of SHH signaling, why do you change to Ptch mutants, which are SMO inhibitor sensitive?
Line 159 – “radiation resistance by glioma” could authors fix this?
Line 168- “recent studies showed that Hh signaling was also associated with tumor immunity. Patients could have elevated response to Hh inhibitors if Hh mediated tumor immunity is better exploited and targeted” This sentence need to be clarified. What do authors mean with “tumor immunity”?
Line 172 – It would be nice if authors define some acronyms: “PD1, CTLA-4, TIM3, LAG3, TIGIT, and IL-10 in exhausted T cell as well as PD-L1/2, CD80/86, OX40L, CD137L, IDO, and CCL22” Explain something about each of these molecules and their role in cancer….
Line 173- After 173 lines talking about the role of SHH signaling in cancer, authors say: “Mutations in Hh pathway drive life threatening cancers like pancreatic cancer, breast cancer, and MB among others”. Really? This belongs to line 1.
Author Response
This review manuscript by Gampala and Yang try to summarize the recent findings in the role that SHH signaling plays in controlling tumor microenvironment. Unfortunately, I feel that the present topic is poorly reviewed and I don’t feel the manuscript should be accepted in the present form. Here I list a few of issues I found.
Line 37- Define CNS
Response: CNS is now defined.
Line 38- “Despite a decent survival rate, the disease remission rate is very high”. This sentence makes no sense.
Response: We thank the reviewer for pointing out this. The sentence has been removed.
Line 39 – “Research on 100 most influential MB studies indicates that hedgehog (Hh) pathway aberration is the main culprit for MB initiation and progression” This is only true for SHH MB subtype (30% of the cases). The cell of origin of the rest of subtypes is not SHH driven.
Response: We thank for the reviewer’s point. The sentence has been modified as “SHH-MB initiation and progression”.
Line 46- In general, Hh should be named SHH or HH when referring a protein/pathway. People in medulloblastoma field like to call it SHH and not HH. Sometimes you do Shh, other SHH, other Hh… please use one. Shh/Hh is a mouse gene.
Response: We thank the reviewer’s careful examination. Corrections have been made as shown with track changes.
Line 60 – “Deregulation of Hh pathway in cancers includes overexpression of Hh ligand in tumor microenvironment (TME), loss of function mutation in PTCH1 and SUFU, and gain of function mutation in SMO” Should include GLI2 amplifications.
Response: We thank the reviewer for this suggestion. “GLI2 amplifications” has been included in the sentence.
Line 63 – Can authors explain what is type II here: “type II non-canonical Shh signaling…”? I never heard of type II SHH signaling.
Response: We have deleted the type II.
Line 65 – “Brain tumors consist of tumors,” I guess you are missing the word cells after tumors, and should be singular in this case.
Response: We thank the reviewer’s careful examination. Corrections have been made.
Line 67- ‘The SHH enriched MB significantly increases gene expression of tumor-associated macrophages. Tumor- associated macrophages (M2 subtype) are abundantly present in SHH MBs.” You are saying two times the same. Just one sentence would be enough.
Response: We thank the reviewer’s careful examination. The sentence has been modified to avoid confusion.
Line 72 – “GLI can also be activated by upstream regulators, such as TGFb1, BET proteins, PRMTs, and HDAC proteins” BET and HDAC proteins are required for the transcription of Gli1. They are not activators of the pathway, but downstream regulators.
Response: The sentence has been corrected.
Line 73 – [53,54]: These manuscripts are not the original ones describing the regulation of the gli1 promoter by Brd4. The original ones were published years before. PMID: 25355313, PMID: 24973920
Response: Suggested references have been added.
Fig 1 legend: “ON-OFF signaling of the Hh pathway” I don’t see any “on” “off”. Please do dual panels showing the on and the off state. You are only showing the on.
Response: We thank the reviewer’s careful examination. We apologize for this error. Figure 1 has been now modified and detailed description added to reflect the figure.
Line 88 – “Newer methods of targeting such as 88 immunotherapy and stem cell transplantation are emerging but are quite expensive to afford [56]”. The reference is not the right one. Also, can authors explain how a SHH tumor would be cured by using stem cel transplantation…. ?
Response: We appreciate the reviewer’s question. Stem cell transplantation is currently being tested to replace the cells that have tumor causing mutations by using healthy cells from the same individual. Also, bone marrow or peripheral stem cell transplantation is performed that allows for higher doses of chemotherapy drugs to be administered. We have also included appropriate references.
Line 90 – “Current clinical inhibitors against Hh signaling include 5E1 (against Shh); Cyclopamine, Vismodegib, Sonidegib (against 90 SMO); ATO, Gant-58, Gant-61 (against GLI1 transcription factor) among others.” The only clinical inhibitors from this list are vismodegib, sonidegib and ATO. The rest are pre-clinical. Authors seem not to know there is a big difference.
Response: We thank the reviewer’s careful examination. We apologize for this error. The mistake has been corrected.
Line 112 – “Adverse events like hair loss, fatigue, taste alterations appear in most Hh inhibitor treated patients that leads to discontinuation of treatment in at least 30% of patients, but so far, no treatment related deaths have been reported in clinical trial”. You start the GLI inhibitor section by talking about adverse effects in the clinic to GLI inhibitors??? I think you mean SMO inhibitors. If so this text doesn’t belong here. Remember besides ATO, no GLI inhibitor was tested in clinical trials.
Response: We thank the reviewer’s careful examination. The start of this section indeed talks about inhibitors against HH signaling initiators like SMO that exist in clinical trials. The side effects associated with these emphasize the need for identifying alternate therapeutic targets and drugs thus bringing the downstream effectors like GLI1 into picture.
Line 114- “Downstream inhibition of Hh signaling makes it possible to in the event of failure of upstream inhibition.” I cannot understand this sentence. Please fix.
Response: This sentence has been fixed.
Line 122- “A direct GLI inhibitor, Arsenic trioxide (ATO), is the only one that has been approved by the FDA for treating acute promyelocytic leukemia (77)” In the referenced paper GLI in never mentioned. ATO is FDA approved for this type of leukemia, but not as a GLI inhibitor. They even have a table in the manuscript with all tentative ATO targets, and GLI or SHH are not included.
Response: ATO is a well-studied inhibitor of GLI1 and in this sentence we indicated that as well as the fact that ATO has been FDA approved for AML. We apologize for any confusion and also included additional references for ATO inhibition of GLI1.
Line 127/8- “This case revealed various SMO gene mutations in Vismodegib-resistant MB. PTCH1 has also been reported to be mutated thus resulting in Hh activation in majority of BCC and MB.” These 2 sentences make no sense. If you are talking about downstream of SMO activation of SHH signaling, why do you change to Ptch mutants, which are SMO inhibitor sensitive?
Response: The authors apologize for this mistake and the section has been fixed.
Line 159 – “radiation resistance by glioma” could authors fix this?
Response: This has been fixed as indicated by track changes.
Line 168- “recent studies showed that Hh signaling was also associated with tumor immunity. Patients could have elevated response to Hh inhibitors if Hh mediated tumor immunity is better exploited and targeted” This sentence need to be clarified. What do authors mean with “tumor immunity”?
Response: The authors thank the reviewer for raising this question. We changed tumor immunity to tumor immunosuppression which we meant.
Line 172 – It would be nice if authors define some acronyms: “PD1, CTLA-4, TIM3, LAG3, TIGIT, and IL-10 in exhausted T cell as well as PD-L1/2, CD80/86, OX40L, CD137L, IDO, and CCL22” Explain something about each of these molecules and their role in cancer….
Response: We thank for the reviewer’s suggestion. These immune checkpoint modulators, cytokines and chemokines have been discussed extensively in many reviews, and we like to focus on the application of those immune checkpoint inhibitors and HH inhibitors in clinic. Simply pointing out the names of the immune checkpoint modulators will be easier for the reader to catch the point in current review.
Line 173- After 173 lines talking about the role of SHH signaling in cancer, authors say: “Mutations in Hh pathway drive life threatening cancers like pancreatic cancer, breast cancer, and MB among others”. Really? This belongs to line 1.
Response: The authors thank the reviewer for pointing out this. We removed the sentence from this section.
Round 2
Reviewer 1 Report
The authors have taken my comments into account. I consider that the revised manuscript can be accepted for publication.
Author Response
Thanks to the reviewer for your satisfaction with our response.
Reviewer 4 Report
The manuscript still needs additional work. A lot of concepts are still misleading and show not enough knowledge to write a review in the field. Besides, authors ignored many of previous suggestions made to the manuscript, making the overall review process longer than necessary.
Line 18- HH (this is still not fixed in general).
Line 19- Human gene is capital (this happens all over the text).
Line 39 – Again: “Research on 100 most influential MB studies indicates that hedgehog (HhHH) pathway aberration is the main culprit for MB initiation and progression” This is only true for SHH subtype (this was already pointed out in first revision of this manuscript)
Line 56- Has been SMO defined already?
Line 67- “It was also reported that SHH signaling activity alone is not sufficient for advanced development of MB. “ author should explain this concept better, and reference manuscript.
Figure 1: In the top part, GLI target genes: Cell cycle and Cell growth are pretty much same concept. I would pick one.
Figure 1: Botton panel. It is not true that BCC and MB are only driven by PTCH mutations (again: human genes are capitalized). Authors could distinguish between ligand-dependent and ligand-independent tumors instead of the misleading “Ptch1 mutations” and “SHH overexpression”.
Line 95: When I previously asked to explain better how stem cell transplantation could be considered a method to target a tumor I did not mean to add the reference to a clinical trial. The way this is written in misleading. The stem cell transplantation allows the use of higher doses of chemotherapeutics. It is not meant to target the tumor, but to reduce the detrimental effects of the high doses of chemo. This should be either explained and correctly referenced, or totally removed from the manuscript.
Line 98 -Gant 61 is a GLI1 and GLi2 inhibitor (already pointed out in my previous revision).
Line 98- Why authors say: “These molecules have been extensively reviewed and are hence are not described in details in this current review” if they are indeed reviewing most of these drugs in the next paragraphs of the manuscript?
Line 116- “Among other SMO antagonists, Cyclopamine has been used in preclinical studies… “ Cyclopamine is not “another SMO inhibitor”. It was the very first HH inhibitor ever described.
Line 118- Glasdegib is FDA approved for AML since 2018.
Line 119- CK1a agonists and BET inhibitors should be listed here as GLI inhibitors.
Line 120- I already brought this up in my previous review (I guess my comments were partially ignored) I do not understand what this sentence has to be with GLI inhibitors “Adverse events like hair loss, fatigue, taste alterations appear in most HhHH inhibitor treated patients that leads to discontinuation of treatment in at least 30% of patients, but so far, no treatment related deaths have been reported in clinical trials75,76.” There is a lack of flow in the text. These lines could be used when introducing all HH inhibitors, as this concept does not belong to GII inhibitors perse.
Line 123- “Vismodegib-resistant MB exhibits hyperactivation of GLI1 protein and activity.” Activation of SHH signaling downstream of SMO is not unique to MB. Change MB for cancer, and add reference in BCC for example.
Line 130- Again my previous comment has been totally ignored: A direct GLI inhibitor, Arsenic trioxide (ATO)82,83, is the only one that has been approved by the FDA for treating acute promyelocytic leukemia84.” Although ATO inhibits GLI, it is not approved by FDA as a GLI inhibitor. This is misleading.
Line 153- ”PTCH1 mutations” How can Ptch1 mutants go under “Resistance mechanisms to in HhHH inhibitors”. These mutants are not resistant to SMO inhibitors.
Line 161- “SUFU mutations” why SUFU mutants are SMO inhibitor resistant is not explained. What’s the point again of having these mutants under “Resistance mechanisms to in HhHH inhibitors” if this is not explained?
Line 176- Could authors better explain how “drug holiday” could help in preventing treatment resistance? The way it is explained is misleading.
Line 179- “For example, a series of BRD4 inhibitors based on AbbVie's phase I clinical pan-BET inhibitor 2 (ABBV-075), yielded Compound 25 to be a safe, tolerant, and high potent GLI inhibitor both in vivo and in vitro 55 98”. This should be better explained. As BET inhibitors are known to block GLI signaling since 2016, I don’t see how manuscripts published later on could be re-proposing a BET inhibitor for the treatment of SMO inhibitor resistant tumors. Also, reference 98 might be wrong here.
Line 180- “Interestingly, recent studies showed that HhHH signaling was also associated with tumor immunosuppressionnity.” No references here. Authors bring up concepts and have no references in general all over the manuscript.
Table 1- JQ-1 should not be listed as a PD-L1 inhibitor in based to cited publication. Although BET activity might control PD-L1 expression in tumor cells, effect of BRD inhibitors is too broad to be listed here. Remember many papers show BET inhibition blocks GLI, so the effect of JQ-1 on PD-L1 expression might be GLI driven indeed.
Line 265- “ Emerging evidence show that immunotherapeutic strategies can contribute to better cancer patient survival. Pediatric MB is the most frequently diagnosed brain tumor in children. The prognosis is mainly dependent on the molecular subtype of the tumor, although therapeutic strategies are limited to conventional radiation therapy, chemotherapy, and surgery. Due to severe neurological side effects, strategies such as immunotherapy are being actively investigated.” Not a single reference.
Line 256- The conclusion section should be a paragraph or two.
Author Response
Reviewer 4 Response to Comments:
The manuscript still needs additional work. A lot of concepts are still misleading and show not enough knowledge to write a review in the field. Besides, authors ignored many of previous suggestions made to the manuscript, making the overall review process longer than necessary.
Line 18- HH (this is still not fixed in general).
Response: The authors apologize for missing some. All HH protein is represented correctly now.
Line 19- Human gene is capital (this happens all over the text). – Human gene is all Capital but I did not use the gene. I only used pathway. So I did not italicize HH but left it capital non-italicized.
Response: The authors apologize for this and corrected all genes and protein symbols appropriately.
Line 39 – Again: “Research on 100 most influential MB studies indicates that hedgehog (HhHH) pathway aberration is the main culprit for MB initiation and progression” This is only true for SHH subtype (this was already pointed out in first revision of this manuscript) – Don’t know how to rectify this.
Response: We thank the reviewer for pointing this. We corrected it as “Research on 100 most influential MB studies indicates that sonic hedgehog (SHH) pathway aberration is the main culprit for SHH-MB initiation and progression” according to the reviewer’s suggestion.
Line 56- Has been SMO defined already?
Response: We thank the reviewer for pointing out this. SMO has been defined in the corrected manuscript.
Line 67- “It was also reported that SHH signaling activity alone is not sufficient for advanced development of MB. “ author should explain this concept better, and reference manuscript. – Will try to do this.
Response: We thank the reviewer for suggesting this. The sentence has been modified to explain it better as follows “Additional to SHH signaling activity, multiple tumor interacting and influencing cells like vascular cells, immune cells, astrocytes, microglia, stem cells and even extracellular matrix can contribute to advanced development of MB47.”
Figure 1: In the top part, GLI target genes: Cell cycle and Cell growth are pretty much same concept. I would pick one.
Response: Figure 1 has been modified and the section mention by the reviewer has been removed here. However, this change has been made in Figure 2 right top panel. Thank you.
Figure 1: Botton panel. It is not true that BCC and MB are only driven by PTCH mutations (again: human genes are capitalized). Authors could distinguish between ligand-dependent and ligand-independent tumors instead of the misleading “Ptch1 mutations” and “SHH overexpression”.
Response: This change has been made according to the reviewer’s suggestion.
Line 95: When I previously asked to explain better how stem cell transplantation could be considered a method to target a tumor I did not mean to add the reference to a clinical trial. The way this is written in misleading. The stem cell transplantation allows the use of higher doses of chemotherapeutics. It is not meant to target the tumor, but to reduce the detrimental effects of the high doses of chemo. This should be either explained and correctly referenced, or totally removed from the manuscript.
Response: This section has been edited to explain the concept better. We thank the reviewer’s suggestion.
Line 98 -Gant 61 is a GLI1 and GLi2 inhibitor (already pointed out in my previous revision).
Response: This change has been made according to the reviewer’s suggestion.
Line 98- Why authors say: “These molecules have been extensively reviewed and are hence are not described in details in this current review” if they are indeed reviewing most of these drugs in the next paragraphs of the manuscript?
Response: This sentence is now modified to appropriately reflect the following section. We thank for the reviewer’s suggestion.
Line 116- “Among other SMO antagonists, Cyclopamine has been used in preclinical studies… “ Cyclopamine is not “another SMO inhibitor”. It was the very first HH inhibitor ever described.
Response: This sentence is now modified to reflect the correction suggested by the reviewer.
Line 118- Glasdegib is FDA approved for AML since 2018.
Response: We thank the reviewer for pointing this out. We are reporting the current clinical status of Glasdegib for medulloblastoma.
Line 119- CK1a agonists and BET inhibitors should be listed here as GLI inhibitors.
Response: From the literature (see below), CK1a and BET act different mechanism to regulate GLI1, and so far, the CK1a agonists and BET inhibitors are not reported to inhibit GLI1 directly. Therefore, listed as direct GLI1 inhibitors here will be confused to the reader.
CK1a agonist references :
https://cancerres.aacrjournals.org/content/canres/early/2014/08/16/0008-5472.CAN-14-0317.full.pdf
https://biosignaling.biomedcentral.com/articles/10.1186/s12964-018-0236-z
BET inhibitors references :
https://www.ncbi.nlm.nih.gov/pmc/articles/PMC7443367/
https://www.sciencedirect.com/science/article/pii/S221138352030633X
https://www.nature.com/articles/s41388-021-01783-9
https://link.springer.com/article/10.1007/s00277-021-04602-z
Line 120- I already brought this up in my previous review (I guess my comments were partially ignored) I do not understand what this sentence has to be with GLI inhibitors “Adverse events like hair loss, fatigue, taste alterations appear in most HhHH inhibitor treated patients that leads to discontinuation of treatment in at least 30% of patients, but so far, no treatment related deaths have been reported in clinical trials75,76.” There is a lack of flow in the text. These lines could be used when introducing all HH inhibitors, as this concept does not belong to GII inhibitors perse.
Response: This sentence has been moved to the start of the section as advised by the reviewer.
Line 123- “Vismodegib-resistant MB exhibits hyperactivation of GLI1 protein and activity.” Activation of SHH signaling downstream of SMO is not unique to MB. Change MB for cancer, and add reference in BCC for example.
Response: We thank the reviewer’s suggestion. The change has been made and reference was added.
Line 130- Again my previous comment has been totally ignored: A direct GLI inhibitor, Arsenic trioxide (ATO)82,83, is the only one that has been approved by the FDA for treating acute promyelocytic leukemia84.” Although ATO inhibits GLI, it is not approved by FDA as a GLI inhibitor. This is misleading.
Response: This sentence has been removed from this section.
Line 153- ”PTCH1 mutations” How can Ptch1 mutants go under “Resistance mechanisms to in HhHH inhibitors”. These mutants are not resistant to SMO inhibitors.
Response: We thank the reviewer for commenting on this. This section has been removed.
Line 161- “SUFU mutations” why SUFU mutants are SMO inhibitor resistant is not explained. What’s the point again of having these mutants under “Resistance mechanisms to in HhHH inhibitors” if this is not explained?
Response: We thank the reviewer for commenting on this. This section has been removed.
Line 176- Could authors better explain how “drug holiday” could help in preventing treatment resistance? The way it is explained is misleading.
Response: We thank the reviewer for pointing this. Drug holiday helps combat serious side effects of chemotherapy but is not applied for preventing treatment resistance. The two sentences are different in the section – one explains resistance and the other about side effects.
Line 179- “For example, a series of BRD4 inhibitors based on AbbVie's phase I clinical pan-BET inhibitor 2 (ABBV-075), yielded Compound 25 to be a safe, tolerant, and high potent GLI inhibitor both in vivo and in vitro 55 98”. This should be better explained. As BET inhibitors are known to block GLI signaling since 2016, I don’t see how manuscripts published later on could be re-proposing a BET inhibitor for the treatment of SMO inhibitor resistant tumors. Also, reference 98 might be wrong here. – I can try to do this.
Response: We thank the reviewer for pointing this. Even though BET inhibitors are early on reported to block GLI signaling, we still report it as drug re-purposing because: it is reported in referenced literature as is and as its binds to bromodomain containing proteins indirectly effects GLI proteins and not directly.
Line 180- “Interestingly, recent studies showed that HhHH signaling was also associated with tumor immunosuppressionnity.” No references here. Authors bring up concepts and have no references in general all over the manuscript.
Response: We thank the reviewer for pointing this. Reference has been added.
Table 1- JQ-1 should not be listed as a PD-L1 inhibitor in based to cited publication. Although BET activity might control PD-L1 expression in tumor cells, effect of BRD inhibitors is too broad to be listed here. Remember many papers show BET inhibition blocks GLI, so the effect of JQ-1 on PD-L1 expression might be GLI driven indeed.
Response: We thank the reviewer’s suggestion. JQ-1 has been removed from the table.
Line 265- “ Emerging evidence show that immunotherapeutic strategies can contribute to better cancer patient survival. Pediatric MB is the most frequently diagnosed brain tumor in children. The prognosis is mainly dependent on the molecular subtype of the tumor, although therapeutic strategies are limited to conventional radiation therapy, chemotherapy, and surgery. Due to severe neurological side effects, strategies such as immunotherapy are being actively investigated.” Not a single reference.
Response: We thank the reviewer for commenting on this. New references are added at appropriate sentences.
Line 256- The conclusion section should be a paragraph or two.
Response: Somehow, we can’t understand the question. Is it possible to point out specifically in which conclusion section?
Round 3
Reviewer 4 Report
Nothing to add.